# Cellular and Molecular Evidence of the Synergistic Antitumour Effects of Hydroxytyrosol and Metformin in Prostate Cancer

**DOI:** 10.3390/ijms26031341

**Published:** 2025-02-05

**Authors:** Francisco Porcel-Pastrana, Antonio J. Montero-Hidalgo, Miguel E. G-García, Ignacio Gil-Duque, Antonio Prats-Escribano, Manuel D. Gahete, André Sarmento-Cabral, Raúl M. Luque, Antonio J. León-González

**Affiliations:** 1Maimonides Institute for Biomedical Research of Córdoba (IMIBIC), 14004 Córdoba, Spain; porcelpastrana.francisco@gmail.com (F.P.-P.); antonio.montero.96@gmail.com (A.J.M.-H.); miedu1998gargar@gmail.com (M.E.G.-G.); igilduque@gmail.com (I.G.-D.); antoniopratsescribano@gmail.com (A.P.-E.); bc2gaorm@uco.es (M.D.G.); amsbcabral@gmail.com (A.S.-C.); 2Department of Cell Biology, Physiology and Immunology, University of Córdoba, 14014 Córdoba, Spain; 3Hospital Universitario Reina Sofía (HURS), 14004 Córdoba, Spain; 4Centro de Investigación Biomédica en Red de Fisiopatología de la Obesidad y Nutrición (CIBERobn), 28019 Madrid, Spain; 5Department of Pharmacology, School of Pharmacy, University of Seville, 41012 Seville, Spain

**Keywords:** prostate cancer, metformin, hydroxytyrosol, anti-tumour actions

## Abstract

Prostate cancer (PCa) is the tumour pathology with the second highest incidence among men worldwide. PCa is strongly influenced by obesity (OB), which increases its aggressiveness. Hence, some metabolic drugs like metformin have emerged as potential anti-tumour agents against several endocrine-related cancers. Likewise, a high adherence to the Mediterranean diet has been associated with lower rates of OB and a reduction in PCa aggressiveness since this diet contains phenolic bioactive compounds such as hydroxytyrosol (HT) that is mainly present in extra virgin olive oil. Thus, we decided to analyse the therapeutic potential of the combination of HT + metformin in different PCa cell models. Specifically, combinations of different doses of HT and metformin were evaluated by analysing the proliferation rate of LNCaP, 22Rv1, DU-145, and PC−3 cells using the SynergicFinder method. The results revealed a synergistic effect of HT + metformin in significantly reducing proliferation, especially in LNCaP cells. This anti-tumour effect of HT + metformin was also confirmed in migration and tumoursphere formation assays in LNCaP. The effects on the cell cycle and apoptosis were also assessed by flow-cytometry, and a cycle arrest in the G1 phase and an increase in late apoptosis were observed with the combination of HT + metformin. The phosphorylation levels of critical components of different oncogenic pathways were measured which revealed that the combination of HT + metformin significantly reduced the activity of multiple components of the MAPK, AKT, and TGF-β pathways. Overall, the combination of HT + metformin might represent a new therapeutic avenue for the management of PCa patients, an observation that certainly warrants further investigation through a well-designed clinical trial.

## 1. Introduction

Prostate cancer (PCa) constitutes the second most frequently diagnosed malignancy and the fifth leading cause of cancer-related mortality among males globally [1]. While the initial diagnosis often identifies tumours with low risk of progression, it is concerning that 30–50% of these cases may progress to advanced PCa [especially to the phenotype of Castration-Resistant Prostate Cancer (CRPC)] within a 5–10-year period, which remains an incurable and ultimately fatal illness [2]. Therefore, patients who develop this advanced PCa require a transition to active therapeutic interventions [3]. Due to the typically indolent progression of PCa, there is a significant interest in identifying therapeutic options with minimal adverse effects and ideally, treatments that are efficacious in all phenotypes of PCa. An emerging approach to treat several cancer types is based on the repositioning of drugs that are already approved to treat other pathologies since this strategy is faster and easier to translate to clinical practice compared to the development and validation of new drugs [4]. In this context, one of the most promising research lines in cancer research focuses on drugs commonly used for treating certain metabolic disorders (e.g., type 2 diabetes, obesity, and hypercholesterolemia), such as metformin or statins (e.g., simvastatin) [5,6,7,8,9,10,11,12,13]. In fact, several epidemiological studies have reported that patients treated with metformin showed a lower risk of developing various malignancies, including PCa [13,14,15,16], and that metformin in combination with androgen deprivation therapy (ADT) could decrease some of the unfavourable metabolic consequences of ADT and enhance the tumour-suppressive effect of ADT [17,18].

Within this framework, numerous investigations have also explored the potential of bioactive compounds derived from edible fruits and vegetables within the Mediterranean diet for the treatment and prevention of PCa [19,20]. Notably, there is a burgeoning interest in bioactive compounds present in extra virgin olive oil (EVOO), particularly phenolic alcohols [e.g., tyrosol and hydroxytyrosol (HT)] and their derivatives (e.g., oleocanthal and other secoiridoids), in many oncological contexts [21]. Among these bioactive compounds, HT has demonstrated promising anticancer properties on different in vitro and in vivo cancer models primarily due to its function as an antioxidant [13,22,23,24,25,26,27]. In fact, our group previously demonstrated the anti-tumour effects of HT and its chemical derivatives on PCa cells [28].

Interestingly, independent researchers have pinpointed synergic anti-tumour effects of the combination of different bioactive antioxidant compounds with metformin in various cancer types [29]. However, to the best of our knowledge, the potential anti-tumour effects of the combination of HT and metformin in PCa cells remains largely unknown. For that reason, this study was conceived to pursue the following objective: to explore the in vitro anti-tumour effects of the combination of metformin and HT in different human PCa cell models [androgen-sensitive (LNCaP and 22Rv1), androgen-independent (DU-145), and CRPC (PC-3) models] to analyse the ability of these drugs and their combination to alter the activity of multiple key oncogenic signalling pathways that are critical for tumour development, progression, and aggressiveness in human PCa.

## 2. Results

### 2.1. Synergic Effect of Hydroxytyrosol and Metformin in Prostate Cancer Cell Models

To determine the potential interaction effect of the combined treatment with HT and metformin, we performed a dose–response matrix with five different concentrations (see Table 1) based on previous results from our group [11,16,28]. To this end, we focused on the inhibition of cell proliferation as a functional readout to identify interaction effects for every combination. We selected LNCaP and 22Rv1 cells as androgen-sensitive cell models, DU-145 cells as an androgen-independent cell model, and PC−3 cells as a CRPC cell model. Based on the IC50 analyses, LNCaP cells were the most sensitive to both treatments with the highest Combinatory Synergy Score (based on SynergyFinder software 2.0 [30]), followed by 22Rv1 cells (both the LNCaP and 22Rv1 cell models showed synergy effects), while DU-145 and PC−3 cells had the lowest Synergy Scores, showing an additive effect (Table 2). It should be mentioned that the IC50 could not be obtained for metformin or HT in PC−3 cells since a proliferative effect lower than 50% was obtained in response to all the doses used (Table 2).

More specifically, LNCaP cells were the cell model showing the most effective actions in response to the individual or combination treatments (HT and/or metformin; Figure 1A; two-way ANOVA statistics shown in Table 3; Dunnett’s multiple comparison test results shown in Appendix A−Table A1). As mentioned above, LNCaP cells also showed a stronger synergy effect (calculated using the Synergy Combination Index; Table 2). For the 22Rv1 cell model, the synergic effect was also evident but was milder than that observed in the LNCaP model (Table 2), with the main effect induced by HT (Figure 1B; two-way ANOVA statistics shown in Table 3; Dunnett’s multiple comparison test results shown in Appendix A−Table A1). The proliferation rate was reduced in DU-145 cells only with the maximum doses of both treatments (HT and metformin), but no overall significant interaction was found between them (Figure 1C; two-way ANOVA statistics shown in Table 3; Dunnett’s multiple comparison test results shown in Appendix A−Table A1). Finally, the proliferation rate in PC−3 cells was slightly, but significantly, reduced in response to HT or metformin using different doses and showed a low inhibitory interaction with the maximum doses of both treatments (Figure 1D; two-way ANOVA statistics shown in Table 3; Dunnett’s multiple comparison test results shown in Appendix A−Table A1). Based on these results, we selected LNCaP cells (an androgen-sensitive model) for the following analyses since it showed the highest sensitivity to the combination of HT and metformin (Table 2 and Table 3, and Figure 1). From all the combinations, 10 μM HT and 2.5 mM metformin were selected to further explore the synergic effects of these treatments because this was the combination that was able to exert a prominent synergic effect on the proliferation rate and it reduced the needed dose for both compounds when used individually.

### 2.2. Hydroxytyrosol and Metformin Synergistically Reduce Aggressiveness Parameters of Prostate Cancer Cells

To further explore the synergic effect of HT + metformin in PCa cells, we treated LNCaP cells with HT (10 μM) and metformin (2.5 mM), alone or in combination, as well as with the control vehicle for 24 h and performed migration and tumoursphere growth assays. Both individual treatments significantly reduced the migration rate of LNCaP cells (Figure 2A). Interestingly, the combination of HT + metformin potentiated the effect of the individual treatments, reducing the migration rate by more than 50% compared to the vehicle-treated cells (Figure 2A). In the tumoursphere growth assay, we observed a similar behaviour but, in this case, only the combination of HT + metformin significantly reduced this functional parameter in LNCaP cells (Figure 2B). Individually, HT and metformin moderately reduced tumoursphere growth but these reductions did not reach statistical significance (Figure 2B).

Thus, to further explore the cellular response to HT and metformin, we decided to perform apoptosis (early and late apoptosis, performed at 6 h and 48 h after treatment, respectively) and cell cycle (by flow cytometry) assays. In the apoptosis assay, after 6 h of the treatments (early apoptosis), there were no differences relative to the control vehicle (Figure 2C). In contrast, when we evaluated late apoptosis induction (treatments incubated for 48 h), metformin alone and the combination of HT + metformin were able to induce apoptosis, but the combination appeared to show a higher rate of apoptosis stimulation than metformin alone, although this difference was not significant (Figure 2D). Next, we found that the combination of HT and metformin provoked a significant increase in cell arrest at the G1 phase of the cell cycle compared to the control and individual treatments in LNCaP cells, resulting in a significant reduction in the proportion of cells in the S/M phase compared to the control and individual HT treatments, and in the G2 phase compared to the individual HT treatment (Figure 2E). Metformin alone was also able to provoke cell arrest at the G1 phase compared to the vehicle-treated cells. Altogether, our in vitro results confirmed the potential functional anti-tumour effects of the combination of HT and metformin in the LNCaP cell model.

### 2.3. Combination of Hydroxytyrosol and Metformin Altered Relevant Molecular Pathways in Prostate Cancer Cells

We used a phosphorylation array to interrogate the potential molecular mechanisms underpinning the functional anti-tumour effect of the combination of HT + metformin in the LNCaP cell model. With this approach, we simultaneously evaluated the phosphorylation of 47 proteins implicated in four different oncogenic pathways (specifically the MAPK, AKT/mTOR, JAK/STAT, and TGF-b pathways; Figure 3A). In general, we found that the combination HT + metformin and the individual treatments might exert an anti-tumour effect through the down-phosphorylation of these signalling pathways (Figure 3B). Specifically, the main effectors of the MAPK signalling pathway, ERK1 and ERK 2, showed a down-phosphorylation in response to the combination of HT + metformin compared to the individual treatments, while other factors (i.e., MKK6 and CREB) showed a similar downregulation in response to the combination and individual treatments (Figure 3C). Additionally, the metformin treatment seemed to be more effective in altering the AKT/mTOR signalling pathway, as we observed stronger inhibitory effects of the metformin treatment alone compared to the combination treatment of HT + metformin (Figure 3B,D). However, the phosphorylation of some upstream (i.e., PDK1) and downstream (i.e., BAD, GSK3b, 4E-BP1, and p70S6K) factors were more downregulated in response to the combination treatment of HT + metformin compared to the individual treatments (Figure 3D).

Furthermore, we found that the JAK/STAT signalling pathway was not drastically altered in response to the HT and/or metformin treatments, and only EGFR was slightly down-phosphorylated in response to the combined HT + metformin treatment compared to the individual treatments (Figure 3E), while some proteins of the STAT family (i.e., Stat2 and Stat3) were slightly over-phosphorylated (Figure 3E). Remarkably, our results revealed that the main inhibitory effects of the combined treatment with HT + metformin seemed to be associated with the TGF-b pathway, since this combination was able to markedly reduce the phosphorylation levels of ATF2, SMAD1, SMAD2, and SMAD5 (key components in the phosphorylation cascade), and also c-Fos and c-Jun (transcription factors directly related to proliferation, cell cycle arrest, and apoptosis) compared to the individual treatments (Figure 3F). Moreover, given the relevance of the AR signalling pathway in PCa pathophysiology, we decided to measure both *AR* mRNA levels (Figure 3G) and *AR* activity (using the signature described by Spatt. et al. [31]; Figure 3H) in response to HT and/or metformin but no differences were found in the treated vs. non-treated (control) cells.

## 3. Discussion

PCa is a significant global health concern due to its widespread occurrence and high death rate [1]. Unfortunately, the current treatments (surgery, hormone therapy, radiotherapy, and chemotherapy) fail for many patients [3]. This leads to a decline in quality of life for the patients and their families, especially for those diagnosed at later stages, and generates substantial healthcare costs, thereby reinforcing the necessity to find novel therapeutic approaches that can be fast-tracked into the clinic. An emerging therapeutic approach in the oncology field is repositioning drugs that are already approved for other pathologies [4]. In this scenario, recent studies from our group have demonstrated that drugs used to treat metabolic disorders (such as metformin), as well as bioactive compounds within the Mediterranean diet [i.e., those derived from edible vegetables or present in extra virgin olive oil (EVOO), such as hydroxytyrosol (HT; the main phenolic antioxidant in EVOO)], exert anti-tumour actions in PCa [13,14,15,16,28,32,33]. These separate findings, together with the potential benefits of combining metformin with natural antioxidants reported by other groups [29], prompted us to evaluate whether the combination of metformin and HT could translate into strong beneficial anti-tumour actions in PCa cells compared to the individual treatments. Therefore, to the best of our knowledge, the present report provides the first comprehensive analysis of a direct side-by-side comparison of the anti-tumour effects that metformin, HT, and the combination of both compounds exert on different human PCa cell models [androgen-sensitive (LNCaP and 22Rv1 cells), androgen-independent (DU 145 cells), and CRPC (PC−3 cells) models], and of their possible underlying mechanisms (i.e., modulation of the activity of multiple key oncogenic cellular processes and signalling pathways that are critical for tumour development, progression, and aggressiveness in human PCa).

First, we performed a dose–response analysis based on a proliferation assay and the SynergicFinder method [30] in the LNCaP, 22Rv1, DU 145, and PC−3 cell models. This approach revealed that LNCaP cells were the cell model exhibiting the highest sensitivity (synergistic response) and a significant interaction effect in response to the combination of HT and metformin. Moreover, and as previously reported, treatment with HT alone mainly affected 22Rv1 cells [28], while metformin alone affected all the PCa cell lines [11,16]. Remarkably, we demonstrated that the anti-proliferative effect of the co-treatment with HT + metformin was significantly more pronounced than the individual treatments with HT or metformin (used at the same concentration as in the combination) in LNCaP cells. In line with these data, we also demonstrated that the combination of HT and metformin enhanced the observed anti-migratory effect as well as the reduction in tumoursphere growth compared to the individual treatments in LNCaP cells. Importantly, it should be emphasized that among all the established human PCa cell lines, LNCaP cells are unique in their ability to model key aspects of PCa progression, including AR signalling and responses to therapeutic agents [34,35]. Therefore, these overall results might be therapeutically relevant and suggest that a well-designed clinical trial is warranted to investigate the effects of the combination of HT + metformin to reduce key clinical aggressiveness features in PCa patients. In line with this, previous studies have also proven that the combination therapy of metformin and different plant-based bioactive compounds or even with other therapies (i.e., chemotherapy, radiotherapy, targeted therapy, and immunotherapy) might exert more significant anti-tumour and/or clinical beneficial effects than monotherapy in different cancer types [36,37,38,39,40,41,42]. However, to the best of our knowledge, no other studies have reported similar patterns of responses to the combination of metformin and EVOO bioactive compounds, specifically HT, in PCa or in other endocrine-related cancers. Therefore, these findings emphasize the innovative nature of this approach and highlight the need for further research to evaluate the potential utility of these drugs, particularly their combination, as an effective therapeutic tool for the management of different human tumour pathologies, including PCa.

Next, to further explore the cellular outcomes that underlie the synergistic functional effects observed in response to the combination of HT and metformin in PCa cells, we evaluated two functional and molecular endpoints, apoptosis and cell cycle arrest, that are critical in the pathophysiology of cancer cells. This approach revealed that the combination of HT + metformin increased the late apoptosis of LNCaP cells due to cell cycle arrest. Thus, these findings suggest that the proposed drug combination of HT + metformin holds potential clinical relevance in PCa cells with characteristics similar to those of LNCaP cells, which demonstrated significant sensitivity in our study. However, further studies are necessary to evaluate the applicability of these findings in other PCa subtypes to better understand its broader clinical potential [43]. In support of these findings, previous studies have reported a similar enhancement effect with the combination of metformin and other bioactive antioxidant compounds (i.e., resveratrol) in other cancer cells, including breast cancer [MCF-7 [44] and MDA-MB-23 [45]) and pancreatic carcinoma [46]. Furthermore, it has been also reported that the combination of metformin with curcumin (a diarylheptanoid from the roots of turmeric (Curcuma longa)) is able to increase apoptosis in LNCaP cells, but cell cycle arrest was not affected by this combination [47]. Overall, our data demonstrated that the combination therapy of HT + metformin not only exerted a significantly more pronounced increase in apoptosis through cell cycle arrest, but also significantly reduced proliferation, migration, and tumoursphere formation compared to the individual treatments in LNCaP cells, which should be potentially considered as a novel anti-tumoural avenue to treat PCa patients.

Finally, to interrogate the signalling mechanisms underlying the synergistic anti-tumour actions of the combination of HT and metformin in PCa cells, we analysed the activity of different oncogenic signalling pathways that are important in PCa pathophysiology. Interestingly, we found that the combination of HT + metformin altered the phosphorylation pattern of different critical signalling pathways in PCa cells in a more pronounced way than the individual treatments with HT or metformin, including a further down-phosphorylation of ERK1 and ERK2 (from the MAPK pathway), P70S6K and 4E-BP1 (from the AKT pathway), as well as c-Fos, c-Jun, ATF2, and SMAD1/5 (from the TGF-b pathway). Modulation of the ERK pathway has been associated with PCa progression [48] and it has been widely reported that the inhibition of ERK activity can prevent tumour growth, induce apoptosis and cell cycle arrest, and reduce metastasis in PCa [48,49,50]; similar effects were observed in other cancer types such as breast [51] and bladder cancer [52]. Likewise, P70S6K and 4E-BP1 are effectors of mTOR and an increase in their activity is associated with the up-regulation of the transcription of genes involved in enhancing cell proliferation and tumourigenicity [53]. The inhibition of P70S6K has been demonstrated to reduce cell proliferation, directly affecting the ability of tumours to grow and maintain themselves [54]. Inhibition of 4E-BP1 induces an anti-tumour response in prostate [55,56] and colorectal cancer [57]. Furthermore, c-Fos and c-Jun modulation is involved in the development of aggressive phenotypes of PCa (from an androgen-dependent to androgen-independent phenotype) [58], wherein a reduction in c-Jun activation has been related to a lower migrative phenotype, which is associated with regulation from SMAD proteins [59]. Additionally, both HT and the combination of HT + metformin reduced the phosphorylation levels of SMAD2, which is a regulator of c-Jun in the canonical TGF-b pathway in PCa cells and is correlated with a poorer prognosis [60]. On the other hand, a general degradation of ATF2 is associated with the suppression of the aggressive features of PCa [61]. Therefore, these data provide original, compelling evidence that the combination of HT + metformin is functionally linked to these well-known, patho-physiologically relevant, oncogenic pathways in PCa (the MASPK, AKT, and TGF-b pathways). Most of the changes observed in the activity of these oncogenic signalling cascades could explain the aforementioned synergic anti-tumour actions of the combination of HT + metformin in terms of increasing apoptosis and inhibiting proliferation, migration, and tumoursphere formation compared to the individual treatments in LNCaP cells. Finally, we did not find any significant changes in the expression or activity of AR (a key oncogenic factor associated with PCa pathophysiology and linked to several of the dysregulated signalling pathways observed in this study [62]) in response to HT and/or metformin. It should be mentioned that previous studies have reported modulation of AR expression in response to metformin; these divergent observations between our study and previous studies might be explained in part by the short exposure time used in our study compared with other studies [63].

In conclusion, this study provides solid evidence indicating that the combination of HT + metformin synergistically reduces the aggressiveness features of androgen-sensitive PCa cells (i.e., decrease in proliferation rate, migration capacity, and/or tumoursphere formation, and activation of apoptosis and cell cycle arrest) compared with the individual treatments with both drugs; these actions are likely mediated by a more pronounced alteration of distinct key oncogenic signalling pathways (i.e., the MAPK, AKT, and TGF-b pathways) that may result in these synergistic effects. Although further work will be required to complete our understanding of this cellular process and to fully elucidate the translational potential behind these interesting and potentially relevant observations, this study suggests the use of these drugs, especially their combination, as a potential new therapeutic tool for the management of PCa that should be tested soon for their use in humans through a well-designed clinical trial.

## 4. Materials and Methods

### 4.1. Cell Cultures and Reagents

LNCaP, 22Rv1, DU 145, and PC−3 cell lines were obtained from the ATCC (CRL-1740, CRL-2505, HTB-81, and CRL-1435 respectively), cultured, and maintained under the manufacturer’s recommendations, as previously reported [28,32]. Briefly, RPMI 1640 (Corning, Glendale, Arizona, USA; Ref: 15-040-CV) was used and supplemented with 1 % L-glutamine (BIOWEST, Nuaillé, France; Ref: X0550-100), 1% antibiotic–antimycotic solution (Sigma-Aldrich, Madrid, Spain; Ref: A5955-100ML), and 10% FBS (Gibco, Thermo-Fisher, Waltham, MA, USA; Ref: 10270106). The cell lines were validated by analysing short tandem repeats (GenePrint 10 System; Promega, Madison, USA; Ref: B9510) and they were systematically checked for mycoplasma contamination, as previously described [64].

3-Hydroxytyrosol (HT; Ref: 91404) and metformin (Ref: D150969) were obtained from Sigma-Aldrich. Stock solutions were prepared by dissolving the HT powder in dimethyl sulfoxide [DMSO (Applichem, Chicago, USA; Ref: A3672-0100)] and the metformin powder was dissolved in a phospho-buffer solution (PBS). The final DMSO concentration was 0.1% (*v*/*v*). DMSO and PBS were dissolved in culture medium and used as the control vehicle.

### 4.2. Synergy Assay Design and Analysis

SynergyFinder software [30] was used to evaluate the interaction between HT and metformin in PCa cells. To that end, five concentrations (see Table 1) were selected for each reagent based on the effective concentrations that were described in our previous papers [8,11,16,28]. A proliferation assay was performed to evaluate all the possible combinations. Then, the Loewe additivity model [30,65] was used to calculate the Combinatory Synergy Score for every combination (CSS). Following the SynergyFinder instructions, the CSS was interpreted as follows: (i) antagonistic effect when the CSS is < −10; (ii) additive effect when the CSS is between −10 and 10; and (iii) synergistic effect when the CSS is > 10.

### 4.3. Cell Proliferation, Migration, and Tumoursphere Growth Assays

Cell proliferation was evaluated in LNCaP, 22Rv1, DU 145, and PC−3 cells using the resazurin assay (Canvax Biotech, Cordoba, Spain; Ref: CA035S), as previously described [11,66]. Briefly, cells were seeded into 96-well plates at a density of 3000 cells/well. Then, resazurin was added to each well to a final concentration of 10%, followed by 3 h of incubation in a humidified incubator at 37 °C, and then the fluorescence (560/590 nm) was measured using a FlexStation III system (Molecular Devices, Sunnyvale, CA, USA). Cell proliferation was determined at 0 and 48 h after treatments with HT and/or metformin.

Cell migration was evaluated using Millicell Cell Culture Inserts (Sigma Aldrich, Ref: PI8P01250) following the manufacturer’s instructions. Briefly, 30,000 cells pre-treated with HT and/or metformin for 24 h were resuspended in 300 μL of serum-free RPMI 1640 medium and seeded into the transwell inserts, which were placed in the wells of a 24-well plate with 300 μL complete medium (10% FBS). The cells were incubated at 37 °C for 24 h, and the migrated cells were stained and fixated with crystal violet (0.05% crystal violet and 6% glutaraldehyde). Then, after fixation, the crystal violet was recovered from cells using 10% acetic acid, and the absorbance at 560 nm was determined using a spectrophotometer. A negative control with serum-free medium in the lower chamber was used in each experiment.

Tumoursphere formation was also analysed in response to the HT and/or metformin treatments, as previously described [66,67]. Briefly, 2000 cells/well were seeded into Corning Costar 24-well ultra-low attachment plates (Corning; Ref: CLS3473) with DMEM F-12 (Gibco; Ref: CLS3473) medium supplemented with 20 ng/mL EGF (Sigma-Aldrich; Ref: E9644), 10 ng/mL FGF (Peprotech, London, UK; Ref: 100-18B), and B27 (Thermo-Fisher, Waltham, MA, USA; Ref: 12587010). All supplements were replenished every three days. After 10 days, once tumourspheres were formed, the cells were treated with HT and/or metformin. Pictures were taken at 0 and 48 h after treatment. Images were analysed by using FIJI software v1.54 [68]. The tumourspheres’ growth was calculated as a fold change by dividing the area of tumourspheres at 48 h by that at 0 h.

### 4.4. Flow Cytometry (Apoptosis and Cell Cycle Assays)

Flow cytometry was used to analyse the cell cycle and apoptosis features as previously described [8]. Briefly, LNCaP cells were seeded into 6-well plates (∼2 × 105 cells/well) and treated with HT and/or metformin for 6 h (to evaluate early apoptosis) and 48 h (to evaluate late apoptosis). The cells were trypsinized and then washed with PBS. To evaluate the cell cycle, 200,000 cells were fixed with ice-cold 70% ethanol at 4 °C. The FITC Annexin V Apoptosis Detection Kit I (BD Pharmingen, San Jose, CA, USA; Ref: 556547) and 7-aminoactinomycin D (7-AAD, Invitrogen, Waltham, MA, USA; Ref: A1310) in PBS were used to quantify apoptosis and cell cycle arrest, respectively, according to the manufacturer’s instructions. To determine the percentage of apoptotic cells, Annexin V+/Propidium iodide (PI)− cells were designated as early apoptotic cells, whereas Annexin V+/PI + cells were identified as late apoptotic cells. The percentages of cells at the G0/G1, S, and G2/M phases of the cell cycle were calculated by the Watson Pragmatic mode [69]. Cell debris and aggregates were excluded using a sequential gating strategy based on cell size and granularity as previously described [8]. The cells were analysed using a BD LSRFortessa SORP flow cytometer Software v9.0 (BD Biosciences, San Jose, CA, USA; Ref: SCR_018655).

### 4.5. Phosphorylation Array

Protein extracts of LNCaP cells were collected in lysis buffer from 6-well plates after 2 days of treatment with DMSO and PBS dissolved in the culture medium (control), HT (10 μM), metformin (2.5 mM), or HT-Met (selected combination doses). An exploratory analysis of altered signalling pathways was performed by using the Human Phosphorylation Pathway Profiling Array C55 kit, following the manufacturer’s instructions (Raybiotech, Peachtree Corners, GA, USA; Ref: #AAH-PPP-1) and as previously reported [8,28]. Briefly, membranes for the semi-quantitative detection of 55 phosphorylated human proteins belonging to the MAPK, AKT, JAK/STAT, NF-κB, and TGF-β signalling pathways were incubated for 30 min with blocking buffer at 25 °C and then incubated overnight at 4 °C with 1 mL of a 4-fold dilution of the LNCaP cell lysates (*n* = 6; pooled). After washing, the membranes were incubated with a detection antibody cocktail at 25 °C for 2 h and then with a horseradish peroxidase (HRP)-labelled anti-rabbit secondary antibody at 25 °C for an additional 2 h. The signals were collected after adding an ECL reagent using a chemiluminescence detection system (BioRad Universal Hood II, BioRad Laboratories). Densitometric analysis of the array spots was performed using FIJI software [68]. Positive control spots were used as a normalizing factor. The results are expressed as log2 of the fold change of the signal of each protein compared with the signal of the control. The NF-κB results were discarded because of unusable signals.

### 4.6. RNA Isolation, Retrotranscription, qPCR, and AR-Activity Calculation

RNA was isolated from LNCaP cells treated with HT and/or metformin (48 h) using the TRIzol reagent (Thermo Fisher Scientific), as previously described [66]. The RNA was treated using the RNase-Free DNase Kit (QIAGEN, Shanghai, China) to remove DNA. A Nanodrop One Spectrophotometer (Thermo Fisher Scientific) was used to determine the total RNA concentration and purity. cDNA was synthesized from the total RNA using the cDNA First Strand Synthesis Kit (Thermo Fisher Scientific) and random hexamer primers. Real-time quantitative PCR (qPCR) was performed using a CFX Duet Real-Time PCR System (Bio-Rad, Hercules, CA, USA) with the iTaq Universal SYBR Green Supermix (Bio-Rad). Normalization was performed using a normalization factor calculated by GeNorm 3.3 software [70] using *ACTB* and *GAPDH* expression levels, as previously reported [67]. The AR signalling activity (AR-score) was determined as the sum of the ranked expression levels of eight canonical AR-regulated genes that represent markers of AR activity (*ACSL3*, *FKBP5*, *KLK2*, *KLK3*, *NKX3-1*, *PLPP1*, *RAB3B*, and *STEAP1*) [31], as previously described [66]. The primers used in this study can be found in Appendix A−Table A2.

### 4.7. Statistical Analysis

Two-way ANOVA was performed to determine the statistical interaction between HT and Met (results shown in Table 2). All possible comparisons between all the combinatorial doses were performed against the negative control using Dunnett’s multiple comparison test (results shown in Supplemental Table 1). Statistical differences between two independent conditions were determined using unpaired parametric t-tests or nonparametric Mann–Whitney U tests based on the normality, which was assessed using the Kolmogorov–Smirnov test. The assays were performed in at least 3 independent experiments (*n* ≥ 3) and with at least 2 technical replicates. *p* < 0.05 was considered statistically significant. A trend for significance was indicated when the *p*-values ranged between >0.05 and <0.1. In the Human Phosphorylation Pathway Profiling Array, a fold change (log2) of 0.2 was used as the significance threshold. All the analyses were performed using GraphPad 9.4.1 software.

## Figures and Tables

**Figure 1 ijms-26-01341-f001:**
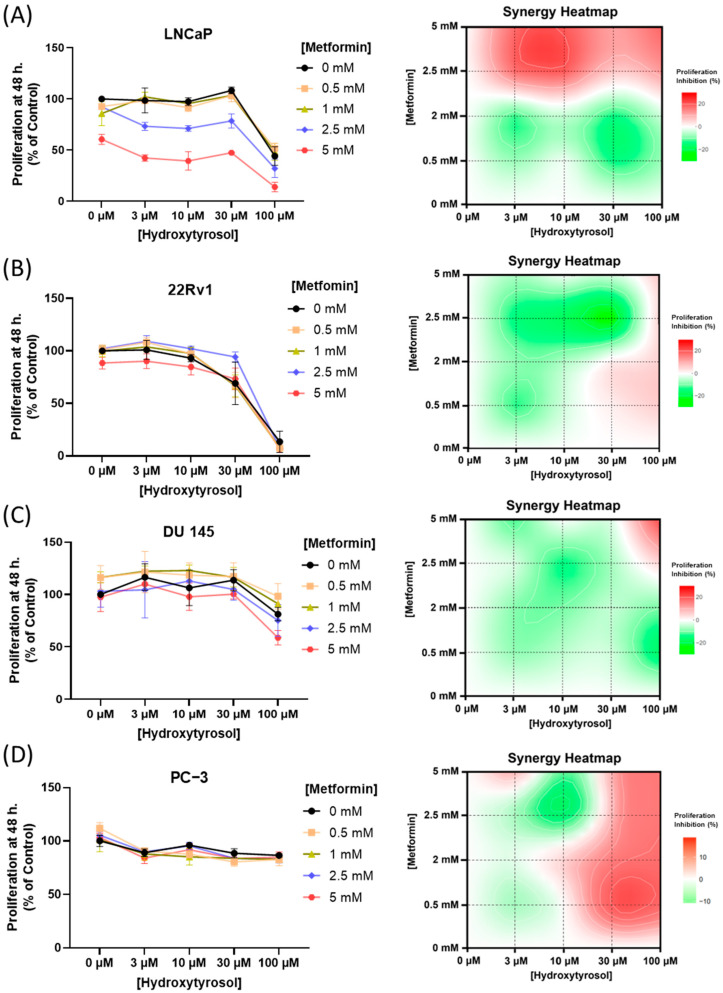
Dose–response curves of hydroxytyrosol and metformin in prostate cancer cell lines. Proliferation rate in response to hydroxytyrosol, metformin, and their different combinations in LNCaP (**A**) left panel), 22Rv1 (**B**) left panel), DU 145 (**C**) left panel), and PC−3 (**D**) left panel) cell lines after 48 h of incubation. Control set at 100%. Data are represented as percentage relative to negative control (mean ± SEM; *n* ≥ 3). Heatmaps of synergic scores generated with SynergyFinder 2.0 for LNCaP ((**A**) right panel), 22Rv1 ((**B**) right panel), DU 145 ((**C**) right panel), and PC−3 ((**D**) right panel) cell lines with proliferation inhibition data.

**Figure 2 ijms-26-01341-f002:**
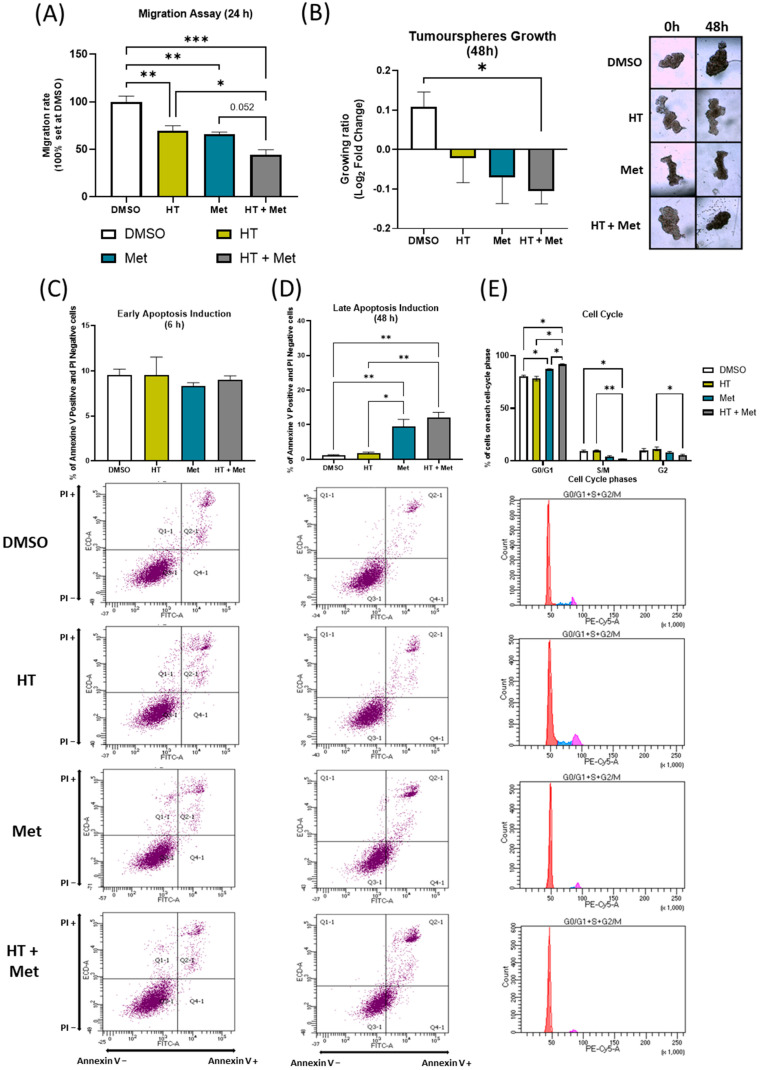
Functional effect of treatment with combination of hydroxytyrosol and metformin. (**A**) Cell migration rate of LNCaP cells treated for 24 h with 10 µM HT, 2.5 mM Met, and their combination. Data are expressed as percentage of migration of control cells (set at 100%) and represent the mean ± SEM (*n* ≥ 3). (**B**) Tumoursphere growth of LNCaP cells after already-formed tumourspheres were treated for 48 h with 10 µM HT, 2.5 mM Met, and their combination. (**C**) Early (at 6 h) and (**D**) late apoptosis (at 48 h) were determined by flow cytometry using Annexin V and PI staining. Data represent cells with positive Annexin V and negative PI staining. (**E**) Percentage of cells in each cell cycle phase, determined using the Watson Pragmatic model. Asterisks indicate significant differences compared to DMSO controls, which were set at 100%. Red represents phase G0/G1, blue represents phase M/S, and purple represents phase G2. Tendency to reach statistical significance is represented by *p*-values and asterisks (* *p* < 0.05, ** *p* < 0.01, and *** *p* < 0.001) indicate statistically significant differences between groups. Abbreviations: HT, hydroxytyrosol; Met, metformin; HT + Met, combination of hydroxytyrosol and metformin.

**Figure 3 ijms-26-01341-f003:**
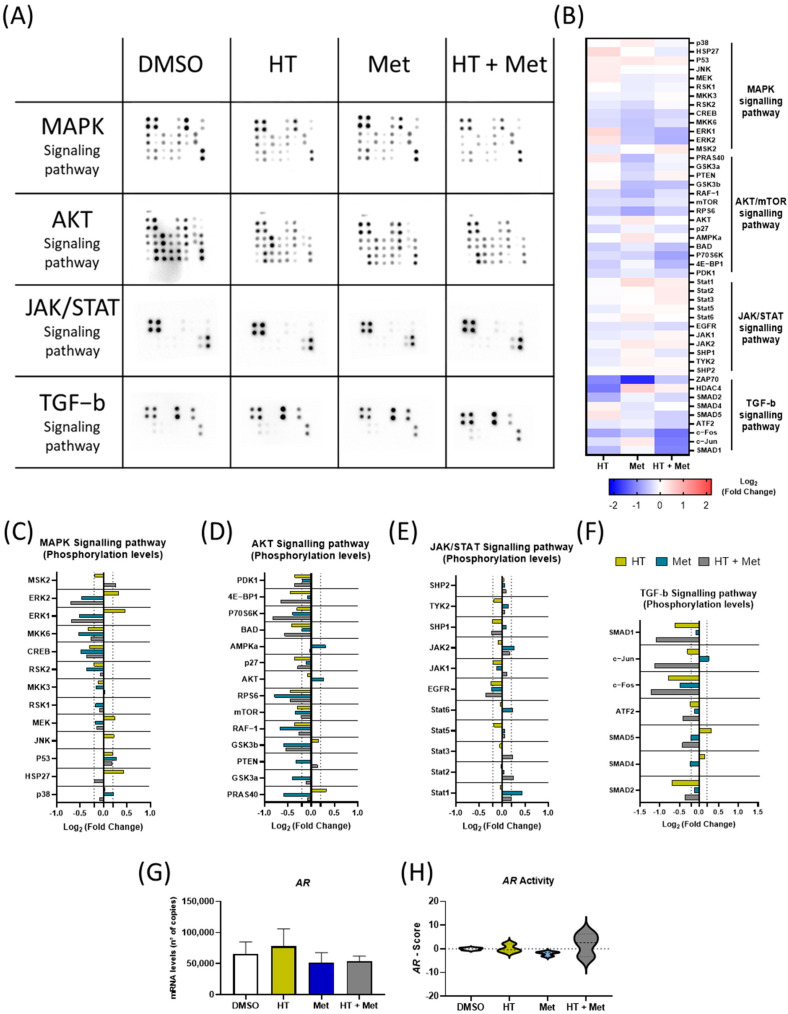
Effect of HT, Met, and their combination on the phosphorylation of key proteins involved cancer-related pathways in LNCaP cells. (**A**) Phosphoprotein array membranes showing the quantified spots reflecting the phosphorylation levels of 47 proteins belonging to four oncogenic pathways (MAPK, AKT, JAK/STAT, and TGF-b) in response to 48 h of treatment with HT, Met, and their combination (*n* = 6; pooled). (**B**) Heatmap showing the mean log2 fold change of the phosphorylation signal corresponding to each depicted protein. ((**C**–**F**)) log2 fold change of phosphorylated protein levels in comparison with the control (DMSO) condition. The dotted line indicates a fold change of ± 0.2. *AR* mRNA levels (**G**) and AR activity (**H**) were measured in treated LNCaP cells. Abbreviations: HT, hydroxytyrosol; Met, metformin; HT + Met, combination of hydroxytyrosol and metformin.

**Table 1 ijms-26-01341-t001:** SynergyFinder 2.0 dose design. Doses were selected based on Refs. [8,11,16,28].

SynergyFinder 2.0 Design					
Drug	Dose 1	Dose 2	Dose 3	Dose 4	Dose 5
Metformin (mM)	0	0.5	1	2.5	5
Hydroxytyrosol (μM)	0	3	10	30	100

**Table 2 ijms-26-01341-t002:** IC50 values for metformin and hydroxytyrosol treatments and Combinatory Synergy Index Scores (based on SynergyFinder software [30]).

Individual Treatment IC_50_				
Drug	LNCaP	22Rv1	DU-145	PC-3
Metformin (mM)	2.313	4.394	5	-
Hydroxytyrosol (μM)	43.66	74.84	86.30	-
Combinatory Synergy Index	52.08	31.16	6.08	1.379
Combination effect	Synergy	Synergy	Additive	Additive

**Table 3 ijms-26-01341-t003:** Two-way ANOVA statistics for drug interaction effects in cell proliferation assay. Asterisks represent statistical differences based on ANOVA among all doses used for each treatment group and the interaction between treatments (ns, *p* > 0.05; *, *p* < 0.05; ****, *p* < 0.0001).

	LNCaP		22Rv1		DU 145		PC-3	
Source of Variation	Adjusted *p* Value	Summary	Adjusted *p* Value	Summary	Adjusted *p* Value	Summary	Adjusted *p* Value	Summary
Interaction	<0.0001	****	0.0284	*	0.9085	ns	0.0016	**
Hydroxytyrosol	<0.0001	****	<0.0001	****	<0.0001	****	<0.0001	****
Metformin	<0.0001	****	<0.0001	****	<0.0001	****	0.0386	*

## Data Availability

Not applicable.

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
