# Peer review of "Cellular and Molecular Evidence of the Synergistic Antitumour Effects of Hydroxytyrosol and Metformin in Prostate Cancer"

_ijms, 2025, doi:10.3390/ijms26031341_

Round 1

Reviewer 1 Report

Comments and Suggestions for Authors The paper entitled "Cellular and molecular evidence of the synergistic antitumor effects of hydroxytyrosol + metformin in prostate cancer", submitted to IJMS by Francisco Porcel-Pastrana et al.  analyses the therapeutic potential of the combination of HT+metformin in PCa cells. Although there is a rationale in support of the  combination of metformin and HT in prostate cancer, the experimental design needs to be improved to increase the overall scientific quality of the manuscript and stenght the conclusions. Here are my major points: 1)LNCaP cells are androgen sensitive and not androgen dependent. The androgen receptor in LNCaP cells contains a mutation in the ligand binding domain which affects steroid binding characteristics and response to antiandrogens. 2)The PC3 cell line should be included in the panel of cell lines tested because is the most widely used model of CRPC 3)The IC50 of single drugs at 48 hours must be calculated (inhibition curve) and provided in the results before testing the various combination. 5)The method used for synergy calculation is unclear. The authors tested increasing concentration of the 2 drugs, without fixing the drugs concentration ratio. The calculation of the CI (combinatorial Index)is missing and mus be provided, otherwise it is not possible to distinguish between an additive and a synergistic effect. The significance of the heatmap provided is unclear and cannot replace the CI calculation. 6)Phospho-arrays were performed only once with a pool of six samples. The authors should repeat the analysis three times with three independent replicates. 7) In the conclusion the Authors claim: “These data are likely relevant clinically and might have a potential translational/oncogenic  implication since the current therapeutic strategies for PCa are not efficient in many cases, especially those with CRPC, where in it has been recently proposed that eradicating cancer cells by triggering apoptosis cell death, as opposed to growth inhibition, should reduce the chances of cancer cell adaptation and therapeutic resistance”. However the results indicate that only LNCap cells respond to the proposed drug combination. The authors should revise their conclusion and highlight the clinical rlevance of thir finding and the real potential application of the proposed combination.  

Author Response

 Point-by-point responses to the Reviewers’ comments

We sincerely acknowledge the Editor and the Reviewers for their constructive comments on our manuscript untitled “Cellular and molecular evidence of the synergistic antitumor effects of hydroxytyrosol + metformin in prostate cancer”, which have been very useful to significantly improve the quality of our study. Accordingly, specific changes have been made in the manuscript, based on these comments, as it is described in detail below in a point-by-point description of the changes introduced, and how. Reviewer’s concerns were addressed. Changes within the manuscript are highlighted in red

We honestly trust that our new results and responses will help to strengthen the support of the Reviewer and hope that this revised version of our manuscript fits within the high-quality scientific standards of the International Journal of Molecular Sciences.

REVIEWER COMMENTS

* Comment 1: LNCaP cells are androgen sensitive and not androgen dependent. The androgen receptor in LNCaP cells contains a mutation in the ligand binding domain which affects steroid binding characteristics and response to antiandrogens.

Author´s Response: We really thank the Reviewer for this relevant indication. Based on the reviewer´s comment, we have specified the androgen sensitive nature of LNCaP across the revised manuscript (lines 81, 93,121, 242, and 337)

* Comment 2: The PC3 cell line should be included in the panel of cell lines tested because is the most widely used model of CRPC.

Author Response: We really thank the Reviewer for this pertinent suggestion. Following the reviewer´s comment, we have also tested the effects of metformin and hydroxytyrosol alone or in combination, and the results are included in a new Figure 1D, table 2, table 3, and supplemental table 1

* Comment 3: The IC50 of single drugs at 48 hours must be calculated (inhibition curve) and provided in the results before testing the various combination.

Author Response: We again really thank the Reviewer for this very appropriate suggestion. Following the reviewer´s request, we have included in a new table 2 providing the IC50 of hydroxytyrosol, metformin, and the combination in all the cell models tested.

* Comment 4: The method used for synergy calculation is unclear. The authors tested increasing concentration of the 2 drugs, without fixing the drugs concentration ratio. The calculation of the CI (combinatorial Index) is missing and must be provided, otherwise it is not possible to distinguish between an additive and a synergistic effect. The significance of the heatmap provided is unclear and cannot replace the CI calculation.

Author Response: We also thank the Reviewer for this pertinent comment/suggestion. It should be mentioned that given the different ranges of effective concentrations of hydroxytyrosol (from 3 to 100 µM) and metformin (from 0.5 to 5 mM), we were not able to fix a single drug concentration ratio. Accordingly, we tested five concentrations based on our previous studies, as it is mentioned in Materials and Methods section 4.2 (Synergy assay design and analysis). Following the Reviewer´s suggestion, the combinatory synergy index has been included in a new table 2 in each of the cell model analysed (i.e. CI of 52.08, 31.16, 6.08, and 1.379 in LNCap, 22Rv1, DU1-45 and PC-3, respectively) and the combinatory additive or synergistic effect found in each of the cell models tested (based on SynergyFinder software). Accordingly, the description of this analysis has been included in the methodology section 4.2 (Synergy assay design and analysis; line 365-373).

* Comment 5: Phospho-arrays were performed only once with a pool of six samples. The authors should repeat the analysis three times with three independent replicates.

Author Response: We sincerely appreciate the Reviewer´s comment. We agree that performing three independent replicates for each condition would possibly represent a more robust approach. However, it should be emphasized that a similar approach has been previously used and accepted in different manuscripts recently published by our group in authoritative journals in the field [For instance: M.C Vázquez-Borrego, et al. A Somatostatin Receptor Subtype-3 (SST3) Peptide Agonist Shows Antitumor Effects in Experimental Models of Nonfunctioning Pituitary Tumors. Clin Cancer Res 2020, doi: 10.1158/1078-0432.CCR-19-2154; A.C. Fuentes-Fayos, et al. Metformin and Simvastatin Exert Additive Antitumour Effects in Glioblastoma via Senescence-State: Clinical and Translational Evidence. EBioMedicine 2023, doi: 10.1016/J.EBIOM.2023.104484; A.J. León-González, et al. Comparative Cytotoxic Activity of Hydroxytyrosol and Its Semisynthetic Lipophilic Derivatives in Prostate Cancer Cells. Antioxidants 2021, doi:10.3390/ANTIOX10091348 M.L. Libero, et al. The Protective Effects of an Aged Black Garlic Water Extract on the Prostate. Nutrients 2024, doi:10.3390/nu16173025]. Based on this fact, and given that this experimental approach would be honestly really expensive (significantly exceeding the budget that the laboratory has to perform this analysis), we would respectfully request to the Reviewer that this additional (and challenging) experiment is not considered as a requisite for the potential acceptance of our present revised manuscript.

* Comment 6:  In the conclusion the Authors claim: “These data are likely relevant clinically and might have a potential translational/oncogenic implication since the current therapeutic strategies for PCa are not efficient in many cases, especially those with CRPC, where in it has been recently proposed that eradicating cancer cells by triggering apoptosis cell death, as opposed to growth inhibition, should reduce the chances of cancer cell adaptation and therapeutic resistance”. However, the results indicate that only LNCap cells respond to the proposed drug combination. The authors should revise their conclusion and highlight the clinical relevance of their finding and the real potential application of the proposed combination.  

Author Response: We really thank the Reviewer for this pertinent suggestion. Following the reviewer´s suggestion, we have tone down our conclusion to clearly state the limitations of our findings while emphasizing the potential relevance of this approach in the specific context of LNCaP-like cell models, as follows (line 280-284]: “these findings suggest that the proposed drug combination of HT + metformin holds potential clinical relevance in PCa cells with characteristics similar to those presented in LNCaP cells, which demonstrated significant sensitivity in our study. However, further studies are necessary to evaluate the applicability of these findings in other PCa subtypes to better understand its broader clinical potential [43]”.

Reviewer 2 Report

Comments and Suggestions for Authors

The manuscript by F. Porcel-Pastrana et al shows interesting data regarding the combined effect of HT and metformin (MET) on human prostate cancer cells.  However, additional information and revisions are needed for this paper to be considered suitable for publication:

1   1. The authors note that the AR positive LNCaP cells are the cell line most sensitive to the HT+ MET combination.  However, they do not include any information regarding the effect of the drug combination on AR protein levels or AR signaling.  Data showing the effect of the combination on expression of AR and its target genes (such as PSA or NKX3.1) should be included since 1) it has been previously shown that metformin regulates AR expression and function and 2) AR has been shown to interact with many of the signaling pathways identified in the phosphorylation array.

2    2. The authors note the combination of HT and MET alter phosphorylation of SMAD1 and SMAD5.  However, the data in panel 3F suggest that the combination also alters phosphorylation of SMAD2, another key component of TGFbeta signaling. It is not clear why the authors do not mention this regulation of SMAD2 under section 2.3.

3    3. In lines 232-234 of the discussion, the authors note that LNCaP cells can model key stages of prostate cancer progression, including tumorigenesis and metastasis.  While LNCaP cells were derived from a metastatic prostate tumor, they cannot be used to model tumorigenesis since they are already malignant cells.  This sentence should be modified to accurately reflect what components of prostate cancer progression can be studied with LNCaP cells. 

Minor Changes:  

Line 22: Modify text so it reads “associated with lower OB and a …”

Lines 163, 181,190 and 305: Fix spelling of TGF-b.

In the panel labels for Figures 3B and 3F fix the spelling of TGF-b.

Lines 333-335:  It is not clear why this sentence is included at the end of the paragraph.  It does not relate to the in vitro studies that were described in the manuscript.

Line 350: Modify text so it reads “crystal violet was recovered from…”

Line 361: Modify text so it reads “growth was calculated as a ..,”

Line 365: Modify text so it reads “~ 2 x 105 cells/well)”

Author Response

Point-by-point responses to the Reviewers’ comments

We sincerely acknowledge the Editor and the Reviewers for their constructive comments on our manuscript untitled “Cellular and molecular evidence of the synergistic antitumor effects of hydroxytyrosol + metformin in prostate cancer”, which have been very useful to significantly improve the quality of our study. Accordingly, specific changes have been made in the manuscript, based on these comments, as it is described in detail below in a point-by-point description of the changes introduced, and how. Reviewer’s concerns were addressed. Changes within the manuscript are highlighted in red

We honestly trust that our new results and responses will help to strengthen the support of the Reviewer and hope that this revised version of our manuscript fits within the high-quality scientific standards of the International Journal of Molecular Sciences.

REVIEWER COMMENTS

* Comment 1: The authors note that the AR positive LNCaP cells are the cell line most sensitive to the HT+ MET combination.  However, they do not include any information regarding the effect of the drug combination on AR protein levels or AR signaling.  Data showing the effect of the combination on expression of AR and its target genes (such as PSA or NKX3.1) should be included since 1) it has been previously shown that metformin regulates AR expression and function and 2) AR has been shown to interact with many of the signaling pathways identified in the phosphorylation array.

Author Response: We really thank the Reviewer for this pertinent suggestion. Accordingly, we have analysed the expression levels of Androgen Receptor (AR) as well as AR-signalling activity [AR-score; determined as a sum of the ranked expression levels of eight canonical AR-regulated genes that represent markers of AR activity (ACSL3, FKBP5, KLK2, KLK3, NKX3-1, PLPP1, RAB3B, and STEAP1) in response to hydroxytyrosol and metformin, alone or in combination, in LNCaP cells. These analyses revealed that neither hydroxytyrosol or metformin alone or even in combination significantly affected AR mRNA levels of AR-signalling activity. These new data and their associated methods and discussion have been included in the revised version of the manuscript [i.e. as new Figures 3G and 3H (see Figures below), and the new section 4.6 of the Material and methods (Lines 433-447: RNA isolation, retrotranscription, qPCR and AR-activity calculation)]. Moreover, primers used in these analyses have been also included in a new Supplemental table 2.

SEE FIGURE ATTACHED IN THE WORD FILE OR FIGURE 3G AND FIGURE 3H IN THE MANUSCRIPT

* Comment 2: The authors note the combination of HT and MET alter phosphorylation of SMAD1 and SMAD5.  However, the data in panel 3F suggest that the combination also alters phosphorylation of SMAD2, another key component of TGFbeta signaling. It is not clear why the authors do not mention this regulation of SMAD2 under section 2.3.

Author Response: We really thank the Reviewer´s for indicating this. Following the reviewer´s comment, we have now mentioned the alteration of SMAD2 as a key component of TGFbeta signalling, as follows: “Remarkably, our results revealed that the main inhibitory effects of the combined treatment with HT + metformin seemed to be associated to the TFG-b pathway, since this combination was able to markedly reduce the phosphorylation levels of ATF2, SMAD1, SMAD2, and SMAD5 (key components in the phosphorylation cascade)…” (lines 203-207), and “Additionally, both HT and the combination of HT + metformin reduced the phosphorylation levels of SMAD2, which is a regulator of c-Jun in the canonical TGF-b pathway in PCa cells and is correlated with a poorer prognosis [60]” (lines 317-320).

* Comment 3: In lines 232-234 of the discussion, the authors note that LNCaP cells can model key stages of prostate cancer progression, including tumorigenesis and metastasis.  While LNCaP cells were derived from a metastatic prostate tumor, they cannot be used to model tumorigenesis since they are already malignant cells. This sentence should be modified to accurately reflect what components of prostate cancer progression can be studied with LNCaP cells. 

Author Response: We sincerely appreciate the Reviewer´s comment. Following the Reviewer´s suggestion, we have modified this sentence as follows (lines 258-261): “Importantly, it should be emphasized that among all the established human PCa cell lines, LNCaP cells are unique in their ability to model key aspects of PCa progression, including AR signalling and responses to therapeutic agents [34,35]”.

* Comment 4: Minor Changes: [Line 22: Modify text so it reads “associated with lower OB and a …”; Lines 163, 181,190 and 305: Fix spelling of TGF-b; In the panel labels for Figures 3B and 3F fix the spelling of TGF-b; Lines 333-335:  It is not clear why this sentence is included at the end of the paragraph.  It does not relate to the in vitro studies that were described in the manuscript; Line 350: Modify text so it reads “crystal violet was recovered from…”; Line 361: Modify text so it reads “growth was calculated as a ..,”; Line 365: Modify text so it reads “~ 2 x 105 cells/well)”]

Author Response: We sincerely appreciate the exhaustive revision of our manuscript. We have corrected all the points raised by the Reviewer.

Round 2

Reviewer 1 Report

Comments and Suggestions for Authors

The authors addressed the most of my criticism. I am satisfied with the review.

Author Response

We really thank the Reviewer for the positive comment

Reviewer 2 Report

Comments and Suggestions for Authors

The authors have addressed most of the concerns noted in the initial manuscript review.  There are still some places in the document (text and figures) where TGFb is incorrectly spelled TFGb.  However this is the only remaining issue that needs to be fixed. 

Author Response

We really thank the reviewer for the comment. Following the Reviewer´s indication, we have corrected the term TFGb in lines 186, 205, 217 and 342.